# Physician perceived barriers and facilitators for self-measured blood pressure monitoring- a qualitative study

**Saahith Gondi[1], Shellie Ellis[2], Mallika Gupta[3], Edward Ellerbeck[2], Kimber Richter[2], Jeffrey Burns[4], Aditi Gupta[3,4]***

**1** Department of Biology, Wake Forest University, Winston-Salem, NC, United States of America, **2** Department of Population Health, University of Kansas Medical Center, Kansas City, KS, United States of America, **3** Division of Nephrology and Hypertension, Department of Internal Medicine, University of Kansas Medical Center, Kansas City, KS, United States of America, **4** Department of Neurology, University of Kansas Medical Center, Kansas City, KS, United States of America

* agupta@kumc.edu

## Abstract

**Data Availability Statement:** All relevant data are within the paper.

### Introduction

Improving hypertension management is a national priority that can decrease morbidity and mortality. Evidence-based hypertension management guidelines advocate self-measured BP (SMBP), but widespread implementation of SMBP is lacking. The purpose of this study was to describe the perspective of primary care physicians (PCPs) on SMBP to identify the barriers and facilitators for implementing SMBP.

### Methods

We collected data from PCPs from a large health system using semi-structured interviews based on the Theoretical Domains Framework (TDF). Responses were recorded, transcribed, and qualitatively analyzed into three overarching TDF domains based on the Behavior Change Wheel (BCW): 1) Motivation 2) Opportunity and 3) Capabilities. The sample size was based on theme saturation.

### Results

All 17 participating PCPs believed that SMBP is a useful, but underutilized tool. Although individual practices varied, most physicians felt that the increased data points from SMBP allowed for better hypertension management. Most felt that overcoming existing barriers would be difficult, but identified several facilitators: physician support of SMBP, the possibility of having other trained health professionals to assist with SMBP and patient education; improving patient engagement and empowerment with SMBP, and the interest of the health system in using technology to improve hypertension management.

**Funding:** The study was supported by the National Institutes of Health K23 AG055666 (AG), R61 AG068483 (AG and JB) and P20GM130423 (SE).

**Competing interests:** The authors have declared that no competing interests exist.

## Conclusion

PCPs believe that SMBP can improve hypertension management. There are numerous barriers and facilitators for implementing SMBP. Successful implementation in clinical practice will require implementation strategies targeted at increasing patient acceptability and reducing physician workload. This may need a radical change in the current methods of managing hypertension.

## Introduction

Hypertension is a major modifiable risk factor for cardiovascular events, stroke, chronic kidney disease, dementia, hospitalizations, and mortality [1–6]. In the US only one in two hypertensive adults have adequately controlled blood pressure (BP) [7, 8]. This is a lost opportunity as intensive BP control, if fully implemented, is estimated to prevent ≈107,500 deaths per year [9]. With these data, improving BP control is a national priority and the Office of the Surgeon General is committed to improving hypertension management [10].

The need to treat hypertension was first realized in the late 1900's [11] and a plethora of guidelines have evolved since. Despite evidence-based guidelines, increased physician awareness, and Medicare expenditures in excess of $50 billion/year, success in reducing BP has declined in recent years [12]. These data indicate an urgent need to reassess current strategies and implement new ones for improving hypertension management.

An important, but often overlooked barrier to effective hypertension management is the widely used practice of relying on in-clinic BP readings. Compared to in-clinic BP monitoring, out-of-office or self-measured blood pressure monitoring (SMBP), sometimes also called home blood pressure monitoring, especially when augmented with telemonitoring and/or counseling [13], is more effective in lowering BP [13–16], cost effective [17, 18], preferred by patients, better at detecting white-coat hypertension and masked hypertension [19], and improves outcomes [20]. Some of these benefits are related to the ability to get an increased number of BP readings which enables better decision making by physicians [21]. National initiatives like the American Medical Association/ American Heart Association's "TargetBP" [22], and the Centers for Disease Control and Prevention (CDC)/ Centers for Medicare and Medicaid Services' (CMS's) "Million Hearts" [23] have been working for years on widespread implementation of SMBP. However, despite being proven effective over 20 years ago [14, 15], being integrated into hypertension management guidelines [7, 24], and being the focus of these prominent national initiatives, SMBP has not been widely implemented in the real world; there remains a disconnect between the evidence-based guidelines and implementation of SMBP in clinical practice. Implementation science is an approach ideally suited to address this disconnect [25].

Implementation science is a theoretically based social science devoted to accelerating the uptake, optimal implementation, and sustained use of evidence-based medicine in clinical practice [25, 26]. By identifying underlying determinants related to adoption and implementation of an evidence-based practice, such as SMBP, implementation science methodology can guide the systematic mapping of evidence-based implementation strategies known to address the particular behavioral determinants faced. These determinants span user motivation, opportunity, and capability to use the evidence-based practice [27, 28] as well as the characteristics of the evidence-based practice itself such as its feasibility and compatibility with the practice environment [29].

Identifying barriers is important to develop successful strategies. Some patient [30, 31] and organization level barriers—cost of home BP cuffs and inability to measure BP accurately [19, 21, 32], poorly aligned incentives, and lack of decision support tools and feedback mechanisms [33–35] have been identified. Since primary care physicians (PCPs) are integral to the management of hypertension, understanding their views on SMBP to manage hypertension is essential to develop and successfully implement strategies for SMBP. Although most physicians report use of SMBP in varying form and extent [36], few have policies and systems in place to ensure appropriate use of SMBP [37]. Engaging PCPs in problem solving around the barriers to SMBP not only improves the strategies but builds credibility of interventions for future dissemination.

To provide the needed context for developing strategies to implement SMBP for improved management of hypertension, we summarize the perspective of PCPs; characterize their knowledge, attitude, beliefs, capabilities, motivation, and practices related to hypertension management and SMBP to identify the barriers and facilitators for SMBP.

## Methods

A preliminary survey regarding SMBP followed by a semi-structured interview was conducted among PCPs to understand the perceived barriers and facilitators for SMBP. This project was approved by the University of Kansas Medical Center Institutional Review Board as a Quality Improvement project. Need for consent was waived under the quality improvement determination. Participation in surveys and interviews was voluntary. No financial or other incentives were offered to the respondents. Respondents could refuse to answer any question at any time. The purpose of the survey and the interview was discussed and verbal permission to record the interview was obtained before each interview from all respondents. No additional information or data from clinical trials on SMBP was offered to the respondents.

### Theoretical domains framework

We used the Theoretical Domains Framework (TDF) [28] as our conceptual model. The TDF consolidates 84 constructs of behavior change derived from psychological and social theories into a single framework to assess implementation problems and inform intervention design. The constructs were organized into 12 validated and two additional domains relevant to PCPs' clinical behavior change [38]. We used the TDF to structure the interview guide, specific questions and probes, and to inform the data analysis and results presentation. We further organized these domains of the TDF within the three overarching categories of Motivation, Opportunity, and Capabilities based on the Behavior Change Wheel (BCW) (**Fig 1**) [27, 28]. An individual's *Motivation* consists of processes that promote and energize the behavior. Physical and social *Opportunity* prompt and allow for the behavior to occur. *Capabilities* refer to an individual's psychological and physical ability to engage in the behavior. The BCW organization of the different determinant domains facilitates understanding of the implications of behavioral changes [27, 28]. We chose the TDF among the popular determinant frameworks based on our substantial experience with the framework [39–42], its comprehensiveness of a full range of potential determinants, its thorough exploration of individual motivation pertinent to individual healthcare providers' prescribing decisions, and its utility in combination with the BCW in implementation strategy selection, with the latter two reasons distinguishing it from other popular frameworks.

**Participants.** We interviewed family medicine and internal medicine PCPs from a large health system. The physicians were approached by the research team (AG) during departmental faculty meetings. The research team introduced the study and invited faculty members to

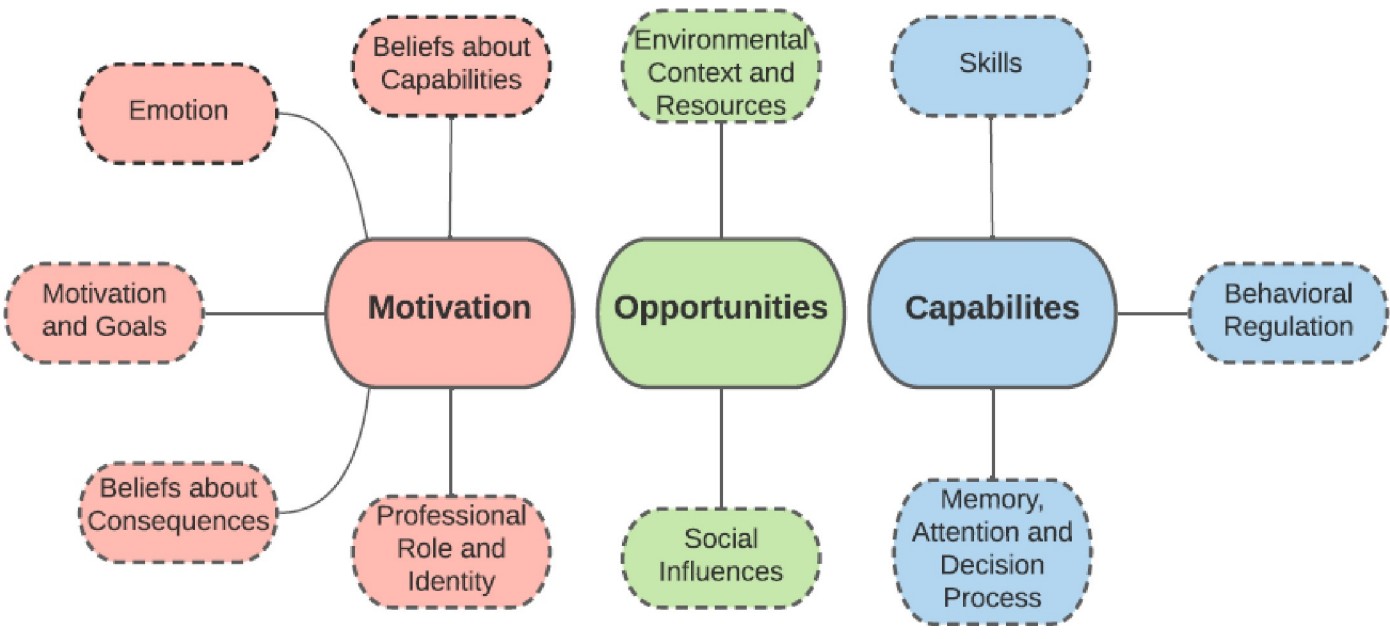

**Fig 1. TDF domains analyzed under behavior change wheel.**

take a brief survey about hypertension management and their use of SMBP with responses in 5-point Likert scale. The participants could access the survey on their phone through a QR code or use a paper form. The last question of the survey asked if they were willing to be contacted for a 30-45-minute interview. The survey was anonymous and only the physicians who agreed to be contacted for the interview were asked to reveal their identity and were later contacted for the interview. Additional interested PCPs were also invited to participate in the interview to achieve thematic saturation.

**Setting.** All respondents were from a single, large health system in the area providing over 920,000 annual outpatient visits and caring for patients from a large geographic area comprised of both urban and rural population. The health systems electronic health record is well developed and has a patient portal (MyChart) actively used by more than 80% of the patients. The patient portal also enables PCPs to obtain self-recorded SMBP readings from patients that can be reviewed by their PCP. The mean age of patients seen by primary care was 57±18 years (2019 pre- COVID-19 pandemic data) with approximately 60% of the patients being female. The racial/ethnic breakdown of patients was 71.8% White, 6% Hispanic, 2.5% Asian, 1.7% Black, and 18% unknown or other. Of the patients with hypertension, 75% were White and 18% Black. Approximately 49% of all patients with a BP reading had at least one systolic BP >130 mmHg. The respondents are expected to have a similar mix of patients as our health system, as most new patients are randomly assigned to available physicians.

**Sample size.** Sample adequacy was assessed by thematic saturation rather than effect size. Based on recommendations to reach saturation [43–45], and our prior experience [46, 47], we anticipated a sample size of about 15 but also planned to increase the sample size if we did not reach saturation on important themes.

**The interview guide.** Two investigators (AG and SE) developed the semi-structured interview guide based on the TDF [48] and our previous qualitiative studies utilizing the approach [39–42]. The interview guide (**S1 File**) explored each domain of the TDF through specific items or prompts and incorporated behaviors including general hypertension management,

SMBP recommendation and management, and potential intervention strategies that might address known barriers to both. Questions were prioritized to ensure a broad range of determinants would be addressed within the participants' time constraints. Consistent with qualitative research, the guide was not considered a verbatim questionnaire, but merely a prompt to ensure the participants considered a comprehensive set of determinants. Because initial survey results indicated some physicians were not enthusiastic about nurse-led SMBP, we specifically probed on the appropriateness of pharmacists' potential role in SMBP. A final question asked the participants to prioritize the determinants discussed.

**Interviewers and data collection.** After identification of interested physicians, a research assistant (SG), a relative "naive" lay person (without previous knowledge of hypertension management or SMBP) conducted semi-structured interviews. The research assistant was trained on use of the interview guide and qualitative interviewing techniques by the qualitative methodologist on the team (SE). The principal investigator (AG), a nephrologist in the same institution with professional relationship with the respondents introduced the research assistant to all the respondents and observed him during the first two interviews. After the initial onboarding period, one-time interviews were conducted independently by the research assistant to optimize explanatory communication behaviors of the PCPs to the "naive" interviewer [49]. We specifically selected a non-physician team member as the interview facilitator to avoid leading or judgmental questions or influencing responses based on clinical knowledge and experience in hypertension management.

Interviews were initially conducted in a private room, usually the physician's office. After March 2020, the remaining interviews were conducted via phone to maintain safety during the COVID-19 pandemic. Based on the physicians' answers, the inclusion and order of the questions varied. The interviews were audio recorded and sent for transcription by a third party.

**Data analysis.** Each transcript was reviewed for mis-transcription and coded using a template analysis strategy by categorizing the individual responses according to the TDF domains. We used a TDF codebook developed *a priori* and used in our previous studies. TDF domain definitions, examples, and counter examples were used for reference to ensure accurate categorization. Interviews were coded by multiple analysts (SG, AG, SE) for training purposes until consensus on the application of TDF domains was reached and completed by a single coder (SG). Interview content assigned to each domain was further synthesized and sub-themes within each were identified. Since most responses fell under 10 domains, two domains; knowledge and nature of behavior; were not reported. The responses were then organized within the three major behavioral determinants based on the BCW [27, 28]. The behavioral differences and trends were noted and general inferences were drawn regarding the overall beliefs. Analytical inferences were drawn regarding the overall beliefs surrounding SMBP utilizing the three BCW intervention drivers. Further analysis regarding SMBP was assessed based on the collective responses and general themes categorized within the three BCW categories to better understand beliefs around future implementation of SMBP.

## Results

Thirty-nine physicians completed the initial survey. Most physicians reported satisfaction with current BP control (**Table 1A**). A majority indicated that additional intervention such as nurses' visits in between PCP visits, more frequent follow up, or use of out of clinic BP monitoring may be appropriate. Physicians were supportive of most of these alternative interventions, although some were less likely to support additional nursing visits. Most physicians recommended SMBP to some extent, but many did not use the patient portal to collect SMBP measurements.

**Table 1. Results of the survey on hypertension management a) from the 39 primary care physicians initially approached and b) from 14 primary care physicians who completed the interview.**

a)

| Question | Responses n (%) | | | | |
|---|---|---|---|---|---|
| | Very satisfied | Somewhat satisfied | Neither satisfied nor unsatisfied | Somewhat unsatisfied | Very unsatisfied |
| **How would you rate your satisfaction with the current BP control of your patients?** | 1 (3) | 21 (54) | 6 (15) | 11 (28) | 0 |
| **Please rate the appropriateness of alternative strategies for improving hypertension management** | | | | | |
| | Very appropriate | Somewhat appropriate | Neither appropriate nor inappropriate | Somewhat inappropriate | Very inappropriate |
| Nurse visits in between provider visits | 27 (69) | 8 (21) | 1 (3) | 1 (3) | 2 (5) |
| More frequent follow up with provider | 16 (41) | 15 (39) | 4 (10) | 4 (10) | 0 |
| Ambulatory BP monitoring | 21 (54) | 13 (33) | 1 (3) | 3 (8) | 1 (3) |
| Home/other out of office (e.g.: grocery store, pharmacy etc.) BP monitoring | 10 (26) | 21 (54) | 5 (13) | 2 (5) | 1 (3) |
| | Always recommend | Often recommend | Sometimes recommend | Rarely recommend | Never recommend |
| **Do you recommend SMBP monitoring to your patients?** | 18 (46) | 18 (46) | 1 (3) | 2 (5) | 0 |
| | Always | Often | Sometimes | Rarely | Never |
| **Do you use the MyChart SMBP monitoring tool?** | 4 (10) | 11 (28) | 7 (18) | 6 (15) | 11 (28) |

b)

| Question | Responses n (%) | | | | |
|---|---|---|---|---|---|
| | Very satisfied | Somewhat satisfied | Neither satisfied nor unsatisfied | Somewhat unsatisfied | Very unsatisfied |
| **How would you rate your satisfaction with the current BP control of your patients?** | 1 (7) | 7 (50) | 3 (21) | 3 (21) | 0 |
| **Please rate the appropriateness of alternative strategies for improving hypertension management** | | | | | |
| | Very appropriate | Somewhat appropriate | Neither appropriate nor inappropriate | Somewhat inappropriate | Very inappropriate |
| Nurse visits in between provider visits | 10 (71) | 3 (21) | 0 | 1 (7) | 0 |
| More frequent follow up with provider | 8 (57) | 3 (21) | 2 (14) | 1 (7) | 0 |
| Ambulatory BP monitoring | 9 (64) | 4 (29) | 1 (7) | 0 | 0 |
| Home/other out of office (e.g.: grocery store, pharmacy etc.) BP monitoring | 3 (21) | 9 (64) | 2 (14) | 0 | 0 |
| | Always recommend | Often recommend | Sometimes recommend | Rarely recommend | Never recommend |
| **Do you recommend SMBP monitoring to your patients?** | 2 (14) | 11 (79) | 1 (7) | 0 | 0 |
| | Always | Often | Sometimes | Rarely | Never |
| **Do you use the MyChart SMBP monitoring tool?** | 0 | 4 (28) | 3 (21) | 3 (21) | 4 (28) |

BP; blood pressure, SMBP; self-measured blood pressure monitoring

Percentages may not add up to 100% across a row due to rounding.

Of the 39 physicians who completed the initial survey, 18 agreed to discuss their experience with hypertension management in more detail. Of these 18 physicians, 14 physicians completed the semi-structured interviews between Dec 2019 and July 2020. These physicians had similar responses to the survey questions for satisfaction with BP control, and use of out of clinic BP monitoring. As adequacy of response did not seem complete for some domains (e.g., knowledge), we recruited three additional physicians to reach saturation at 17. Of the 17

physicians who completed the interview, 15 were women; eight were from the Department of Internal Medicine, and nine from the Department of Family Medicine. Three specialized in geriatrics (one from Internal medicine, and two from Family Medicine). The mean age of the respondents was 47.3±12.2 years, similar to the mean age of all PCPs in the health system (47.6 ±12.6 years). A majority had at least 10 years of clinical experience. Most were White, two were Black and two were Asian, similar to the racial/ ethnic demographics of all primary care physicians in the health system (81.4% White, 7.7% Asian, 4.5% Black, 2.6% two or more races and 3.9% unknow or not listed).

Interviews varied in length and lasted approximately 30–45 minutes. Knowledge, beliefs, and attitudes of physicians towards using SMBP are summarized in Tables 1–3 with direct quotes from the interviews categorized under the three BCW groups, motivation, opportunity, and capabilities.

## Motivation

Within the TDF domain of *Motivation and Goals*, we asked if increasing incentives- financial, work Relative Value Units or clinical time allocated- would increase motivation for SMBP implementation. Physician opinions were mixed. A few physicians were aware of the health systems participation in pay for quality initiatives such as Comprehensive Primary Care Plus (CPC+), a CMS program, that provides financial incentives for improving primary care delivery [50], and felt motivated to use SMBP. Of the physicians who were not aware of these programs, responses regarding the motivating nature of incentives varied greatly. Most indicated that incentives did motivate them. Others felt that these incentives, although helpful, may not be motivating enough to outweigh the logistic problems with SMBP as "barriers will probably still exist" (**Table 2**, motivation and goals). Improvement in hypertension management with SMBP and considering SMBP a part of their duties as a physician served as motivators for some.

Under *Beliefs about Consequences*, physicians believed that SMBP yields more BP readings to manage hypertension, and that SMBP helps diagnose and manage white coat hypertension. These beliefs and consequences reinforced the use of SMBP in the majority of physicians and were facilitators for SMBP (**Table 1**, beliefs about consequences). Despite these benefits, all physicians explicitly believed that SMBP is being underutilized. The most common explanation for underutilization was the barriers outweighing the motivating consequences (**Table 1**, beliefs about capabilities). These barriers, (explicitly addressed under *Opportunity*) created reluctance to use SMBP despite the evidence and motivating consequences.

Within the TDF domain of *Emotion*, most physicians indicated positive emotions for SMBP. Common emotions included feeling hopeful and optimistic about improved care. Physicians indicated that their patients also expressed similar emotions with SMBP. However, some respondents expressed frustration and stress. Although the increased number of BP readings with SMBP helped with improved BP, the overwhelming number of readings and other patient-care related issues that arise with SMBP resulted in these negative emotions (**Table 2**, emotion).

Under *social or professional role and identity*, we explored if other health care workers may be suited to manage or assist with SMBP. Physicians had different degrees of belief in non-physician patient care team members being able to manage SMBP (**Table 2**, social or professional role and identity). Nearly all respondents indicated that hypertension management through SMBP could be assisted, or in some cases performed completely by non-physician patient care team members such as registered nurses, nurse practitioners, and clinical pharmacists. A few physicians felt that non-physician health care workers do not receive adequate training in managing hypertension. These physicians were also motivated to use SMBP themselves, as they perceived SMBP to fall under their job as a physician. However, most felt that with some additional training, assistance from other health care workers would be helpful in SMBP.

**Table 2. Common themes and relevant quotes illustrating barriers and facilitators for *motivation*.**

| TDF Domains | Common Themes noted in Interviews | Exemplary Quotes grouped as Barriers and Facilitators |
|---|---|---|
| Motivation and Goals | Having incentives for SMBP can motivate PCPs. | • . . .all the work that's done in-between office visits currently is not reimbursed, so having some reimbursement for that kind of work would be helpful.—I think one of the things that motivates people and also doctors the most is incentives. |
| | However, some believed incentives alone would not be enough to overcome the barriers. | • So, I don't think physician incentives really play a role. Patient incentives—now, that's a different story.<br>• I think the same barriers will probably still exist though. |
| Beliefs about Consequences | SMBP make more BP readings available to physicians, improving their ability to manage hypertension. | • . . .trying to get their blood pressure down, that's when I'll say hey, why don't you measure your blood pressure readings over five different readings over a period of time with more intensive home blood pressure monitoring.<br>• I feel that sometimes [SMBP] keeps me from adjusting the medicines unnecessarily when their readings in the clinic are high.<br>• . . .I ask them to monitor at home just because we get more data that way. |
| | SMBP is more accurate than in-clinic BP and can help with managing white coat hypertension. | • . . .they're more relaxed at home, and so whenever they see me maybe their blood pressure is a little bit high. . . .<br>• Or some patients have white coat hypertension, which it does help.<br>• I think for white coat hypertension, it's a very helpful tool to have those home readings.<br>• . . .but more optimistic about the information I can get from those readings to be able to make reasonable change for that patient<br>• If it is elevated in the clinic then making a decision about whether or not to start medication and diagnosing hypertension [is hard]. . . |
| Emotion | SMBP is helpful but is stressful and burdensome. | • . . .I'm always hopeful that [SMBP] will be helpful. but I wouldn't say it does not make me feel stressed at all.<br>• I probably feel more hopeful than anything that they'll take action and take a little responsibility for their blood pressure<br>• . . .because it is kind of a burden. |
| Social or Professional Role and Identity | Clinic staff other than physicians can help, but physician involvement is necessary with some aspects of hypertension management. | • I think a team-based approach is definitely the way the future is going. . .the pharmacist for sure, nurse practitioner just fine. RN would be–a trained RN would be fine. And probably a trained LPN would probably be reasonable.<br>• It could be done by a nurse practitioner or pharmacist.<br>• Although they're pharmacists, they would know about interactions probably more than physicians do. But as far as the medical interactions, meaning how is this going to affect their memory loss, how is this going to affect their urination, how is this going to affect their quality of life, how is this going to affect their COPD, how will these medications affect their other illnesses, so I think that's the physician's perspective.<br>• I think there are some things that are specific conditions that you learn in medical school and residency trainings. . .<br>• I think we cannot minimize the role of the physician. I mean after having gone through med school. . .<br>• . . .hypertension does not always have to be an isolated condition. It can be kind of a combination of so many things. Pharmacists I imagine would be able to know maybe medication interaction and see that a medicine is doing that but. . .I don't think you can replace the physician's knowledge.<br>• I think at this point there's enough guidelines and everything that a nurse or a nurse practitioner or a pharmacist would be able to manage . . .as long as there were proper protocols within those guidelines that if it got out of whack, it then resorted back to me. |
| Beliefs about Capabilities | SMBP is underutilized. | • I think we don't get as many people as we would like. So yes, I think it's underutilized. I don't think it's because people are not prescribing it; I think it's because of the barriers for people to do it.<br>• I think we still don't do enough of it. |

*PCP; primary care physician, SMBP; self-measured blood pressure monitoring, COPD; chronic obstructive pulmonary disease.*

**Table 3. Common themes and relevant quotes illustrating barriers and facilitators for *opportunity*.**

| TDF Domains | Common Themes noted in Interviews | Exemplary Quotes grouped as Barriers and Facilitators |
|---|---|---|
| Social Influences | SMBP is driven by evidence and not behavior of colleagues. | • *If there was a study that showed that ambulatory blood pressure monitoring really doesn't help in the long term then I would stop pushing. . .*<br>• *If there was data, it would probably sway me. But I don't know just on other people's actions if I would change.*<br>• *I guess it would depend on why they're using it less and less and if they are aware of some evidence that came out that I'm not aware of. . .*<br>• *Not unless there was some data that showed me that [SMBP] wasn't doing any good.* |
| Environmental Context and Resources | Obtaining SMBP data from some patients can be hard. | • *. . .there's talk about encouraging patients to use the MyChart for uploading blood pressures and how it can be a valuable tool. . . .if the median age is 82, they are going to have more challenges than a younger population to even get access to [MyChart], so that's a big challenge. But I think once someone is on-board, once they have kids that maybe on-board for them, it's a great tool, but it's just giving them that access. . . .there's a push for us to get all of our patients enrolled in MyChart but the older adults just, I get them enrolled and they can't follow through on it.* |
| | Lack of workflow support is a common barrier to SMBP. | • *I'm a pretty good person from a patient standpoint to look at that data and act on it but I don't have that support. . . .*<br>• *I can always use more support but yeah, I feel like I'm kind of old fashioned. I like to look at things myself. Well, no, it might be nice to have more support in that.*<br>• *Obviously if the pharmacist is doing it, it would take some of the burden off of the clinician from having to do it.* |
| | Another barrier is the cost of BP cuffs. | • *If their insurance pays for the blood pressure cuff, I will prescribe it so that they can get it paid for by the insurance company.*<br>• *The cost is all on the patient. . .Insurance isn't going to pay for a blood pressure cuff, and I don't have a free blood pressure cuff that I can give the patient. . .* |
| | Accuracy of home BP cuffs and method of measuring BP is a concern. | *. . . there's a fair amount of worry that the information may be inaccurate because the machines aren't good, or they don't know how to use them and I still have that problem.*• *We just had a patient come in. She was obese. She was using the cuff that was provided with the device and it was too small for her arm, so they were very inaccurate readings. That's why I always have them bring it in, because that stuff happens. . .not being able to always trust what the home values are I guess whether it's 'I'm not sure about the device' or 'I'm not sure how the patient is doing it and that kind of thing' are big barriers.* |
| | Changes to guidelines influence use of SMBP. | • *I would say the more recent guidelines emphasize home blood pressure monitoring, and that has changed my practice*<br>• *Now I think if there was evidence behind [not doing home blood pressure monitoring] then that would sway me.*<br>• *I think the new ones over the last year or two promote home blood pressure monitoring.* |

*PCP; primary care physician, SMBP; self-measured blood pressure monitoring.*

These opinions were closely linked to *beliefs about capabilities*, where physicians felt that SMBP is a useful tool for managing hypertension, but is underutilized due to lack of workflow strategies and other logistical barriers. Assistance from non-physician team members could help increase SMBP utilization and improve patient outcomes.

## Opportunity

Physicians saw an opportunity for SMBP. Under *Social Influences*, most physicians were influenced by published evidence backing SMBP. Only a change in evidence, and not the behavior of their colleagues, would make them not use SMBP (**Table 3**, social influences). Some PCPs found it concerning if others did not use SMBP.

Three main sub-themes were identified under e*nvironmental context and resources*. 1) Patient portals such as MyChart are used to receive SMBP readings from patients. Although all respondents used digital tools for SMBP to a varied extent, a common barrier perceived by all physicians was the technological capabilities of patients. Many felt that older patients have

trouble using these digital tools. Paper logs are a common alternative. When patients can use digital tools, most physicians found patient portals to be helpful. Technical issues with the patient portal varied depending on the preferences of individual respondents, but MyChart was generally well accepted (**Table 3**, environmental context and resources). 2) Another barrier commonly identified was lack of workflow support. The incoming SMBP data can be overwhelming; the number and frequency of readings per patient varied. A common problem was lack of time needed to review and act on the BP readings as there was no specific time allocated for these tasks. Several physicians thought that involving other team members such as registered nurses, nurse practitioners and clinical pharmacists could address this barrier and reduce physician stress (**Table 3**, environmental context and resources). 3) The cost and accuracy of home BP cuffs and the method of measuring BP were also concerns among nearly all respondents (**Table 3**, environmental context and resources). Alternative options to overcome this barrier included obtaining cuffs through programs assisting with free BP cuffs or using pharmacies that offer BP cuff access for the general public. Accuracy of home BP cuffs was also a widely discussed barrier noted by nearly all respondents. Causes of inaccuracy could be related to calibration of the machine, wrist cuffs, improper cuff sizes or technique. Some physicians asked patients to bring home BP cuffs into the clinic to compare readings to detect these inaccuracies (**Table 3**, environmental context and resources).

## Capabilities

Although physician participants reported no personal knowledge deficits for SMBP, they recognized the high skill level necessary for hypertension management, discussed under the *Skills* domain in **Table 4**. Physicians mentioned that knowledge of hypertension guidelines and antihypertensives alone were insufficient to manage hypertension and knowledge about patients' complete medical history, comorbidities and lifestyle were also important (**Table 4**, skills) to understand the cause of hypertension and appropriately manage it.

All respondents stated that ancillary medical staff possess some necessary skills and can be further trained to manage hypertension by themselves or assist physicians (**Table 4**, skills). Nurse practitioners, physician assistants, and pharmacists were specifically discussed. Some physicians indicated that although these professionals may not possess deep knowledge of physiology and disease process- skills acquired in medical school; they can be trained to manage hypertension in most patients. Nurses, nurse practitioners and physician assistants may not have a deep knowledge of anti-hypertensives and hypertension guidelines. Pharmacists on the other hand, may have the knowledge of medications; their doses, side effects and drug interactions, but may not be skilled at a complete clinical assessment or have the expertise needed for considering different aspects of patients' medical problems and lifestyle when considering treatment options.

Under *Behavioral Regulation*, respondents elaborated on differences in using SMBP for managing hypertension. Several mentioned that if their patients are not able to obtain BP cuffs, they ask them to use alternative methods such as pharmacies for recording out of clinic BPs. If transferring SMBP readings digitally is a barrier for patients, some physicians use paper logs to obtain SMBP readings (**Table 4**, behavioral regulation).

The majority of physicians reported different triggers and thresholds for using SMBP in their clinics. These are presented under *Memory, Attention, and Decision Processes* (**Table 4**, memory, attention, and decision processes). Initiation of SMBP for a patient depended on several factors that varied between physicians. Similarly, the threshold for modification of the treatment regimen and the regimen itself also varied. Although the majority followed

**Table 4. Common themes and relevant quotes illustrating barriers and facilitators for *capabilities*.**

| TDF Domains | Common Themes noted in Interviews | Exemplary Quotes grouped as Barriers and Facilitators |
|---|---|---|
| Skills | A holistic approach with knowledge of guidelines, disease processes, medications and comorbidities are needed to manage hypertension. | • *You'd have to have baseline knowledge of the disease and the therapeutic measures, and so you'd have to know the protocols. But you'd also have to be able to think critically, and problem solve, and recognize and understand the exceptions to the rules.*<br>• *I think you have to have some background knowledge of all of those things—the physiology, the pathophysiology, the pharmacology—and then also be able to look at all of those things in the context of the patient and the patient lifestyle and the patient's comorbid conditions so that you can come up with a treatment plan that's going to work for that individual patient.*<br>• *If there is a patient just has plain hypertension and nothing else is going on, that's fine. But if they also have lupus, if they also have a history of a stroke, there are other things to take into account.*<br>• *You would need knowledge of the medications, the guidelines, and yeah, side effects, interactions. There's certainly a large knowledge that's needed to be able to manage somebody's blood pressure.* |
| | Clinic staff other than physicians may not have all the skills needed for managing hypertension but can be trained to manage hypertension. | • *I think that they all have adequate training. The pharmacist may know more about medications. We, in our clinic can now refer to a pharmacist to check in on blood pressure as well as other things, but then we also have the nurse protocol, and so all of them are adequately trained and you could say that a RN nurse can do it.*<br>• *I don't know if they have the skills right now but I think they could be trained. I'm pretty sure they could be trained. We're using people who are PharmDs and some of them have done residencies.. . .I think if a nurse can be taught to change medication based on protocol, that the pharmacist shouldn't have any problem at all.*<br>• *Communication, I guess, is another skill. . . I think there'd have to be a period of training.*<br>• *I think there are some things that are specific conditions that you learn in medical school and residency trainings that a pharmacist just doesn't have the training in so I think there would need to be collaboration and oversight with a provider in certain cases. . .*<br>• *My experience has been that [pharmacists] are knowledgeable about guidelines, medications, side effects. I think the only area that they would not excel in would maybe be looking at the whole picture and knowing the patient individually.*<br>• *I don't know that they could manage all the comorbidities and other things that complicate hypertension and all the other preventative things that need to be going along with it, because it's not their training, but from a strictly medication-data management, side-effects management, they could probably do okay.* |
| Behavioral Regulation | Strategies for SMBP vary among PCPs. | • *I always ask the patient if they have a blood pressure cuff that they use at home, and we talk a little bit about what kind of cuff if they have one. If they have one, we talk about appropriate ways to take the blood pressure. . .*<br>• *Some patients who do not have a cuff and I think are unlikely to purchase one, I either ask them to go to the drugstore and get it checked or come back to our clinic for a nurse visit to have it checked.*<br>• *If my patients are more elderly, they'll usually record them at home and then call my nurse with readings and she'll just type them into a note so I can review them, but a lot of my younger patients are pretty technologically savvy and they'll just use the one on MyChart.*<br>• *But I've had multiple patients who have said that they can't afford it. So, then I often will recommend that they go to a local CVS or Walgreens or a pharmacy and have their blood pressure monitoring.* |
| Memory, Attention, and Decision Processes | PCPs rely on in-clinic BP for hypertension management. | • *I follow the guidelines and everything, so in general 140/90, if it touches that then we start talking about medications and everything.*<br>• *If they come in, though, and their blood pressure is already 180/100, then I'll go straight to medication in addition to the lifestyle modifications.* |

*PCP; primary care physician, SMBP; self-measured blood pressure monitoring, RN; registered nurse.*

guidelines for decision making, management approach was often individualized based on patients' age or other factors (**Table 4**, memory, attention, and decision processes).

## Discussion

We interviewed PCPs, key stakeholders in hypertension management [51], to identify their experiences, beliefs and perceptions toward SMBP. Physicians believed SMBP can improve hypertension management but identified barriers to successful implementation of SMBP. Most believed that it is difficult to adjust anti-hypertensives based on in-clinic BP measurements, that SMBP provides increased number of BP readings, and that SMBP makes it easier to separate reliable BP measurements from situational fluctuations such as white coat hypertension and to determine whether a change in anti-hypertensives is warranted. Despite their beliefs in the strong evidence behind SMBP, there was underutilization of SMBP due to logistical barriers and lack of resources and trained personnel to help with SMBP. They also identified ways to systematically address these barriers to increase adoption of SMBP. Through this study, we aimed to understand physician perspectives so factors inhibiting their use of SMBP can be combined with evidence-based guidelines to develop effective strategies to implement SMBP.

Out of clinic BP, both ambulatory BP and SMBP, are recommended in hypertension guidelines. We focused on SMBP instead of ambulatory BP monitoring as ambulatory BP monitoring is often not well tolerated by patients, more resource intensive, available in specialized clinics only and not easily scalable [19]. Prior studies in [32, 52] or outside the US [53–55] show similar barriers for SMBP despite differences in health policies. Physicians lack appropriate tools and support needed for SMBP, and the use of SMBP remains rather haphazard in clinical practice. Despite similar barriers, use of SMBP is lower in the US- 67% in Spain and 58% in UK compared to 20–50% in the US [32, 56, 57]. In contrast to surveys or questionnaires used in previous studies [32, 52, 53, 55], we used semi-structured interviews to allow for more detailed conversations about the barriers and facilitators. This approach, which is standard in the field of implementation science, allowed us to explore the influence of a comprehensive set of potential determinants, not previously explored in other studies.

Our study identified multiple barriers at different levels with the current systems and policies that prohibit successful implementation of SMBP. Under *motivation* we found that increased workload without additional support staff to train patients on SMBP and follow SMBP readings in an already busy clinic, lack of well-defined protocols for SMBP, lack of detail in the guidelines for SMBP, and cost of BP cuffs were common barriers. New CPT codes 99473 and 99474 introduced in year 2020 might help with physician incentives but are not likely to solve these barriers. Physicians experienced a high degree of stress and perceived burden in managing SMBP. Lack of physician incentives has been identified by other studies but was not a critical barrier in our study. Instead increasing patient engagement and incentives may be more important. Under *opportunities*, the respondents emphasized barriers we attributed to the TDF domain of 'environmental context and resources'. They mentioned that clinical decision support could help- however, no formal algorithm or guidelines are available for widespread implementation. Lack of interdisciplinary teamwork, including training and support of other staff to fulfill the role expectations that physicians believe is necessary was a major barrier. Further, physicians helped highlight barriers at an organizational level. Even though the health system is ideally suited for SMBP, with an excellent electronic medical record system, high patient portal use, physicians who believe in SMBP, largely insured patients, and ongoing institutional initiatives to improve BP control, very few physicians routinely use SMBP (**Table 1**). Incentives by insurers [50] have prioritized improving

hypertension management by health systems and health systems have incorporated tools to facilitate reporting of SMBP. However, our study suggests that financial incentives, while helpful, may only be part of the solution. Few physicians in our study used the tools provided by the health system, suggesting the need for better strategies to monitor SMBP. Our interviews indicated that health system approaches (such as use of a patient portal for entering SMBP) may not be ideal solutions and present additional barriers, particularly for older adults. Under *capabilities* we found that physicians would be more comfortable having additional staff assist with SMBP after they had some additional SMBP-specific training. Prior studies indicate that nurses or pharmacists can manage SMBP, but our study indicated that they might need additional training to fulfil these roles. Patients seen in clinics often have more complex medical issues than participants in clinical trials (where patients with multimorbidity are often excluded), and management of SMBP in the real world may need more training and expertise.

Compared to previous studies using surveys and questionnaires, our interviews also helped us obtain more information on major facilitators for SMBP. Under *motivation* we found that physicians strongly believed in positive outcomes associated with SMBP over current methods to manage BP, and desired to incorporate SMBP in their practice. Physician engagement in SMBP is important as adherence to SMBP by patients is significantly higher if their physicians recommend it [51, 58]. Under *opportunities*, we confirmed that successful implementation of SMBP will require a team approach, efficient data systems, performance assessments, and standardized protocols along with well thought out strategies; all of which are not easy without the support of the organization [59–61]. Under *capabilities* we identified the areas where additional health care professionals can be trained to develop a multi-disciplinary team to implement SMBP. Our study indicates that environmental restructuring and enablement, through adding trained support to care team, physician teams can educate patients on cuff size, the correct method for SMBP, and lifestyle modifications to ultimately improve patient engagement and empowerment [19].

With stakeholders such as patients, physicians, health systems, and insurers interested in adopting SMBP, practical scalable strategies are needed. A systematic implementation science-based approach can integrate evidence-based SMBP in the real world. With the strong evidence backing SMBP, more efforts need to be placed in testing pragmatic effectiveness and implementation studies incorporating education and clear protocols for interpretation and management of SMBP [62].

A limitation of this study is inclusion of PCPs from a single health system. However, this also helped with detailed interviews and meaningful discussions that can lead to actionable steps towards developing an implementation strategy. Moreover, the respondents were a mix of both academic and community physicians from the largest health system in the area. Most of our respondents (88%) were women, limiting generalizability. Our interviews were voluntary and non-incentivized, which could have limited participation. However, the voluntary nature of the interviews enabled participation of physicians committed to hypertension control. Their responses confirmed and expanded on those of other physicians across the world [32, 52–55]. With open-ended questions in the interviews, we were able to identify several facilitators for developing practical strategies to implement SMBP. We did not specifically ask about BP device loaner programs or the new CPT codes 99473 and 99474 introduced in year 2020 to help SMBP. However, none of the physicians mentioned the codes, which might indicate that use of the codes may not be widespread yet.

In conclusion, understanding physician perspective and reasons for their inability to implement SMBP (despite their belief in SMBP) is essential to develop and successfully implement strategies for SMBP. PCPs feel that SMBP is a useful tool to manage hypertension that is underutilized due to lack of workflow, support, patient education and engagement, team

training, protocols, and financing of BP cuffs. These barriers can be addressed by using efficient, user-friendly technology to obtain SMBP readings and developing a collaborative team specifically trained for SMBP. Some of the cost savings in healthcare utilization with improved BP control can be directed towards purchase of BP cuffs by health systems or insurers.

## Supporting information

**S1 File. Identifying barriers and facilitators to home BP monitoring using the theoretical domains framework.**
(DOCX)

## Acknowledgments

We are thankful to all participating physicians.

## Author Contributions

**Conceptualization:** Shellie Ellis, Aditi Gupta.

**Data curation:** Saahith Gondi, Aditi Gupta.

**Formal analysis:** Shellie Ellis, Mallika Gupta, Aditi Gupta.

**Methodology:** Shellie Ellis, Edward Ellerbeck, Kimber Richter, Aditi Gupta.

**Project administration:** Aditi Gupta.

**Resources:** Jeffrey Burns.

**Supervision:** Aditi Gupta.

**Visualization:** Aditi Gupta.

**Writing – original draft:** Saahith Gondi, Mallika Gupta, Aditi Gupta.

**Writing – review & editing:** Shellie Ellis, Edward Ellerbeck, Kimber Richter, Jeffrey Burns, Aditi Gupta.

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
