## [Decision Letter · Decision Letter 0]

17 Mar 2021

PONE-D-20-38491

Physician perceived barriers and facilitators for self-measured blood pressure monitoring- A qualitative study

PLOS ONE

Dear Dr. Gupta,

Thank you for submitting your manuscript to PLOS ONE. After careful consideration, we feel that it has merit but does not fully meet PLOS ONE’s publication criteria as it currently stands. Therefore, we invite you to submit a revised version of the manuscript that addresses the points raised during the review process.

We look forward to receiving your revised manuscript.

Kind regards,

Bradford Dubik

Academic Editor

PLOS ONE

Journal Requirements:

2. Please include additional information regarding the survey or questionnaire used in the study and ensure that you have provided sufficient details that others could replicate the analyses. For instance, if you developed a questionnaire as part of this study and it is not under a copyright more restrictive than CC-BY, please include a copy, in both the original language and English, as Supporting Information. Furthermore, please also provide additional details in the Methods section on the validation of the questionnaire.

When reporting the results of qualitative research, we suggest consulting the COREQ guidelines: http://intqhc.oxfordjournals.org/content/19/6/349. In this case, please consider including more information on the number of interviewers, their training and characteristics; and please provide the interview guide used.”

LAS (Assoc Ed) 09/12/20 ***EO: please send back the following request and ping me with follow-up: “Please amend your current ethics statement to include the full name of the ethics committee/institutional review board(s) that approved your specific study.

3. Please ensure that you refer to Figure 1 in your text as, if accepted, production will need this reference to link the reader to the figure.

Additional Editor Comments (if provided):

Reviewers' comments:

Reviewer's Responses to Questions

**Comments to the Author**

1. Is the manuscript technically sound, and do the data support the conclusions?

Reviewer #1: Yes

Reviewer #2: Yes

2. Has the statistical analysis been performed appropriately and rigorously? 

Reviewer #1: N/A

Reviewer #2: N/A

3. Have the authors made all data underlying the findings in their manuscript fully available?

Reviewer #1: No

Reviewer #2: Yes

4. Is the manuscript presented in an intelligible fashion and written in standard English?

Reviewer #1: Yes

Reviewer #2: Yes

5. Review Comments to the Author

Reviewer #1: This is an interesting study on the perspectives of primary care providers on barriers and facilitators to implementing SMBP. The authors use the theoretical domains framework (TDF) to organize the determinants, which has the potential to inform future interventions. Overall, while interesting, it's not clear that the findings are novel, generalizable or would add to the literature overall. I have included some recommendations for improving the study below.

Overall, the introduction is very well written and makes a good case for the effectiveness and importance of SMBP. It would be helpful to quantify and reference the conclusion that SMBP has not been widely implemented, particularly in the telemedicine era that we find ourselves.

The authors summarize several prior studies on the barriers/facilitators to SMBP implementation. It's not clear how their approach/focus on providers fills gaps in prior research. Have prior studies not examined provider perspectives? Is it clear that provider behavior is key to SMBP implementation?

The authors provide a good definition of the utility of implementation science, but should provide a rationale for their use of TDF/COMB/BCW as opposed to CFIR for example in examining provider based determinants in the methods section.

To really understand context, which is key to implementation science, more information is needed about the setting (urban, rural, large, small, academic, integrated?).

The authors should include the start date of interviews. Did the authors assess pre/post covid changes in perspectives/attitudes?

How was the interview guide developed. There are several examples of TDF interview guides in the literature. Did the authors explore each construct? What kinds of questions were posed of the participants (this should be included in the supplementary materials if possible)?

Reporting should align with COREQ https://www.equator-network.org/reporting-guidelines/coreq/

In the results section, the authors note that many participants did not use the patient portal to collect SMBP measurements. This is again why it's important to understand context. Is SMBP available in your setting? Is the SMBP data integrated into the EHR currently? This is important because if widely available/financed but not utilized, then there may be a way to tie this into prior literature suggesting that SMBP was impossible because of lack of EHR integration, etc.

Were there any differences (e.g., demographic) between individuals who agreed to participate in the interviews and those who did not?

In the motivation section, the interviewers note that they prompted PCPs to answer questions about incentives. Again are these questions that you added to the TDF interview guide?

To what degree did providers receive an introduction on SBMP prior to the interviews (i.e., the fact that nurse assisted SMBP has been shown to be particularly effective).

In the discussion, I found it difficult to tease apart whether/how their findings are different from prior research or how their theoretical approach added to prior literature. The utility of using BCW is that you can map to intervention functions/BCTs. Are they any that the authors can glean that should be tested in future research?

Do the authors have a sense for which one of the TDF constructs were the most salient (e.g, are underuse mostly driven by opportunity or motivation and less so capability?).

Reviewer #2: This manuscript, entitled “Physician perceived barriers and facilitators for self-measured blood pressure monitoring- A qualitative study”, is an important contribution to the literature for self-measured blood pressure monitoring. As the authors note, there is strong science behind the use of SMBP for hypertension management but we have yet to achieve widespread implementation in the U.S. due to a number of patient- and clinician-identified barriers. There are a few publications on barriers but this manuscript helps to shore up physician-identified barriers and facilitators for SMBP implementation.

Overall, this is a well-written piece, but it could benefit from some additional copy-editing (e.g. for missing hyphens in compound adjectives, at least one strange use of a semi-colon). A few copy edits of note:

• Pg 15 of PDF – under “Opportunity” paragraph, third row, delete “expected”.

• Pg 17 of PDF – “Under Behavioral Regulation…” paragraph, third line down, “recoding” should be changed to “recording”

• Table 1 – For the question “Please rate the appropriateness of alternative strategies for improving HTN management”, second to last column heading needs to be changed to “somewhat inappropriate”

• Table 2 – “HPBM” should be changed, at a minimum, to “HBPM” but better to change it to “SMBP” for consistency. (second column, first entry)

Some additional suggestions:

• Recommend editing your third paragraph in the background to mention home-blood pressure monitoring as follows: “Compared to in-clinic BP monitoring, out-of-office or self-measured blood pressure monitoring (SMBP), sometimes known as home blood pressure monitoring, …” I much prefer the term SMBP and much of the currently published literature uses this term but not all of it.

• In your background, you mention the Surgeon General’s support of SMBP (as a strategy from the recent Call to Action to Control Hypertension). Could also consider a sentence acknowledging how national initiatives like AMA/AHA’s TargetBP and CDC/CMS’s Million Hearts have been working for years on widespread implementation of SMBP but have not been able to make much headway.

• For generalizability purposes, this piece could greatly benefit from some additional contextual information about the respondents and their patient populations. For example, the authors provide some written demographic data about respondents (e.g. 15/17 women, 3 geriatricians, mix of both academic and community physicians) but it would be nice to have those data and additional data (age, race/ethnicity) in a table. Would also be nice to understand if any of these 17 key informants were among the physicians who supported SMBP or were they among those who had reservations about using it (from the survey)? Would also be helpful to have some ballpark estimation of what the respondents actual (and perceived if available) BP control rate is for their patient population, what percent of their population has HTN, etc. It would also be helpful to understand a little about the general patient population – e.g. low SES, inner city population; more affluent, mostly-white population. Physician perceptions of barriers and facilitators are skewed by the patients they serve.

• You mention CPC+ but it feels almost like a non sequitur. Consider talking more generally about pay-for-quality initiatives because CPC+ is just one of many initiatives in that space. If you want to call it out, perhaps add a citation. Likewise, you parenthetically mention “A Million Hearts Initiative”. Would recommend changing that to “The Million Hearts Initiative” (or more correctly just “Million Hearts”) and perhaps adding a bit more context or a citation there as well. Million Hearts is a major supporter of SMBP but as an initiative, in general, doesn’t pay for monitors. Were you referencing their work with NACHC/federally qualified health centers or work through departments of public health? Unclear. It is still very much a live initiative (i.e. hasn’t gone away) so additional clarity would be helpful.

• A potential limitation not noted is the fact that almost all respondents were women. While we don’t necessarily have any reason to believe women would have different perceptions about SMBP than men, it is fairly unusual for 88% of respondents in any study to be women and should at least be noted.

• Another potential limitation is, depending on your response to one of my questions above, the respondents were likely all supporters of SMBP so their barriers are already skewed towards people who trust SMBP. Responses would have differed if respondents included physicians who don’t use SMBP with their patients or who don’t trust the readings, etc.

• I would caution you in the future to separate the use of blood pressure kiosks (i.e. found in pharmacies/grocery stores) from SMBP performed on a patient’s own blood pressure monitor. Not a lot of evidence behind the former, lots behind the latter. If you discussed both of them equivalently with physicians in your interviews (the way it was denoted in the survey), that may have skewed their perceptions. May want to list this as a potential limitation.

• Future research is not noted but there seemed to be some gaps in in what was asked of these physicians. For example, what were their feelings about blood pressure device loaner programs (where health care settings bulk purchase devices and loan them out to patients on a temporary basis)? Were they aware of the new CPT codes that were introduced in 2020 that would reimburse clinicians one time to train patients on SMBP technique/perform device calibration (99473) and reimburse physicians up to monthly to receive SMBP readings, interpret, and combine into a patient’s care plan (99474)?

• References 30 and 46 appear to be the same reference.

6. PLOS authors have the option to publish the peer review history of their article (what does this mean?). If published, this will include your full peer review and any attached files.

Reviewer #1: No

Reviewer #2: **Yes: **Hilary K. Wall

---

## [Author Response · Author response to Decision Letter 0]

25 Apr 2021

We thank the editors and the reviewers for their thoughtful feedback and the enthusiasm reflected in their comments. We are also thankful for their suggestions which we believe have improved the manuscript. Our responses to the concerns are outlined below. 

Editors comments: 

1) Please include additional information regarding the survey or questionnaire used in the study. Please also provide additional details in the Methods section on the validation of the questionnaire. Please provide the interview guide used.

Response: Thank you for the comment. We have added the interview guide as supplementary data 1 and added more detail about the interview guide in the Methods section under a new sub section “The interview guide”. Consistent with determinant inquiry which is the current standard in the field of implementation science, we used a qualitative interview guide based on a theoretical framework of behavior change. Rather than using a validated survey instrument designed to elicit a population average from a representative sample of providers, we used a qualitative interview method that seeks to probe all potential determinants of behavior to uncover the range of potential determinants experienced by providers in the targeted intervention setting. This approach ensures that the intervention delivery context is accounted for in implementation strategies and is typically adapted to other contexts as needed during implementation. 

2) Please consider including more information on the number of interviewers, their training, and characteristics 

Response: We have provided additional detail regarding the number of interviewers and described his training (by Dr. Ellis, an experienced ethnographic interviewer), and characteristics under “Interviewers and data collection” in the methods section. We chose a “naive” interviewer to avoid influence of the interviewer in the responses by the “expert” primary care physicians. 

Reviewer #1

1) It will be helpful to quantify and reference the conclusion that SMBP has not been widely implemented, particularly in the telemedicine era that we find ourselves.

Response: We agree. We have added a more recent reference #57 published in September 2020 indicating low use of SMBP. We agree that SMBP should be used more in the telemedicine era and are using telemedicine in our ongoing study on implementation of SMBP (NCT04585880). Our study showed that some of the technological features such as use of a patient portal to report SMBP have their own problems and may not be ideal solutions for SMBP. Although beyond the scope of this manuscript, in our SMBP implementation pragmatic trial (NCT04585880), we are finding that despite data suggesting that telemedicine can increase use of SMBP, barriers to SMBP persist. For example, many patients do not routinely check BP at home, do not report SMBP readings during their telehealth visits and continue to check BP incorrectly. 

2) The authors summarize several prior studies on the barriers/facilitators to SMBP implementation. It's not clear how their approach/focus on providers fills gaps in prior research. Have prior studies not examined provider perspectives? Is it clear that provider behavior is key to SMBP implementation? 

Response: Thank you. We have added some more clarification for findings that are new in our study. Unlike previous studies, we used semi-structured interviews instead of surveys or questionnaires. ‘This approach, which is standard in the field of implementation science, allowed us to explore the influence of a comprehensive set of potential determinants, not previously explored in other studies.’ As indicated in the interview guide (supplementary data 1), we were able to probe deeper into the barriers to SMBP implementation and identify the most critical barriers and their potential solutions. The interviews enabled 2-way conversations and relative advantages and disadvantages of one strategy over another. We were able to identify how strategies such as clinical decision-making tools, physician incentives and having more staff would be helpful. For example, even though pharmacists have shown to be able to manage SMBP, we found that physicians would be more comfortable having pharmacists or nurses assist with SMBP if they had some additional training. This is an important finding for implementing SMBP in clinics as patients might have more complex medical conditions than the selective participants of a clinical trial. We have added more detail in the discussion section to highlight the additional information obtained through this study. 

While we do not believe that provider behavior is the sole cause of poor implementation of SMBP or the only contributor to its successful implementation, PCPs are one of the key stakeholders in implementation of SMBP and ‘understanding physician perspective and reasons for their inability to implement SMBP (despite their belief in SMBP) is essential to develop and successfully implement strategies for SMBP’. Prior studies (ref 51) have suggested that PCPs can greatly influence SMBP. Through our interviews, PCPs provided valuable information on facilitators for SMBP that helped us develop strategies for SMBP implementation.

3) It is unclear if these findings are generalizable.

Response: Qualitative research is not designed to be generalizable. Rather, the in-depth mode of inquiry used here seeks to identify the range of barriers experienced, potential solutions to overcome them, and modalities by which the solutions might be delivered most effectively. Because of the theoretical basis from which the approach is drawn, it is thought to be more efficient at identifying effective solutions which can be systematically adapted as the intervention is brought to scale in other care delivery contexts. 

We included a mix of academic and community clinic physicians, who were trained at different institutions, from different departments and with different number of years in practice, creating a wide range of experience and perspectives across the key characteristics in which we expect the barriers to vary. Moreover, our results are also consistent with some of the previously identified barriers. The responses confirmed and expanded on those of other physicians across the world. From our clinical and research experience, and available data, these results appear to be generalizable, but will be confirmed in our ongoing randomized trial on implementation of SMBP (NCT04585880). 

4) The authors provide a good definition of the utility of implementation science but should provide a rationale for their use of TDF/COMB/BCW as opposed to CFIR for example in examining provider-based determinants in the methods section.

Response: We chose the TDF/COMB/BCW among the popular determinant frameworks based on our substantial experience with the framework, its comprehensiveness of a full range of potential determinants, its thorough exploration of individual motivation pertinent to individual healthcare providers’ prescribing behavior (a construct less emphasized in CFIR), and its utility in combination with the BCW in implementation strategy selection (another feature which is not as robustly developed in CFIR). We have added this in the methods section. 

5) To understand the context, more information is needed about the setting (urban, rural, large, small, academic, integrated?), SMBP availability, SMBP data integration into the EHR. This is important because if widely available/financed but not utilized, then there may be a way to tie this into prior literature suggesting that SMBP was impossible because of lack of EHR integration, etc.

Response: Thank you for this excellent point. We have added a section “Setting” under methods describing the health system and patient population it serves. This section also describes how PCPs can see the SMBP readings entered by the patients on the patient portal. With our interviews we learned that even with this EHR tool, SMBP is low. We have added this under discussion- ‘health system approaches (such as use of patient portal for entering SMBP) may not be ideal solutions and present additional barriers, particularly for older adults. Few physicians in our study used the tools provided by the health system, suggesting need for better strategies to monitor SMBP.’ Like many health systems, we currently do not have an integrated system to directly transfer BP readings from patients cuffs to our EHR. We are however studying this strategy in our current trial. 

6) The authors should include the start date of interviews. Did the authors assess pre/post COVID changes in perspectives/attitudes?

Response: We added the interview dates under the results section. There were no changes in the respondent’s perception. There was however a gap in the interviews due to COVID-19. The study team resumed interviews in June 2020 via a phone vs in person meeting prior to COVID. 

7) How was the interview guide developed? There are several examples of TDF interview guides in the literature. Did the authors explore each construct? What kinds of questions were posed of the participants (this should be included in the supplementary materials if possible)?

Response: We have added a section under ‘Methods’ devoted to the interview guide description, explaining how we developed the interview guide, and the range of domains and behaviors it covered. We have also added the interview guide as Supplementary data 1. 

8) Reporting should align with COREQ https://www.equator-network.org/reporting-guidelines/coreq/

Response: As directed by the COREQ guidelines, we have added more detail about the interviewer, study procedures and analysis. 

Under ‘interviewers and data collection’ section of the methods, we have added personal characteristics of the interviewer, his relationship with the respondents, training, and experience as well as the method of approach. 

Under ‘conceptual framework,’ we have added the rationale for selecting the Theoretical Domains Framework. 

We have added a new section entitled ‘setting’ which describes the institution at which the participants were drawn. 

We have elaborated on the coding procedures in the ‘data analysis’ section. 

9) Were there any differences (e.g., demographic) between individuals who agreed to participate in the interviews and those who did not?

Response: Since the initial survey (which had a question where physicians were asked if they were interested in participating in the interview) was anonymous, we are not able to analyze the difference between the two groups. However, we have added more detail on the respondents’ demographics and age comparison with all PCPs in the health system in the results section. The respondents varied in their clinical experience, age, and apparent race. We could not add the race comparison with the health system due to a lot of missing data. 

10) In the motivation section, the interviewers note that they prompted PCPs to answer questions about incentives. Again, are these questions that you added to the TDF interview guide?

Response: The TDF includes financial incentives as a component of motivation and we routinely explore different intrinsic and extrinsic motivation in our studies guided by the TDF.

11) To what degree did providers receive an introduction on SBMP prior to the interviews (i.e., the fact that nurse assisted SMBP has been shown to be particularly effective).

Response: All our respondents were trained physicians working in an academic medical center. Most of our respondents used SMBP in some form (in a non-standardized way), with some already using nurse assisted SBPM. Our intent was to elicit existing barriers and facilitators to SBPM use, so we did not offer additional education on SMBP. We have added a line in the first paragraph of the methods section to clarify that we did not offer additional education on SMBP. 

12) In the discussion, it was difficult to tease apart whether/how the findings are different from prior research or how their theoretical approach added to prior literature. The utility of using BCW is that you can map to intervention functions/BCTs. Are they any that the authors can glean that should be tested in future research?

Response: We received valuable information based on the BCW. For example, 

a) PCPs find it difficult to manage HTN based on in-clinic BP measurements and indicated their support SMBP. However, they need more resources to implement SMBP. 

b) PCPs identified that financial incentives alone may not be adequate to implement SMBP. 

c) PCPs identified that use of patient portal for SMBP is burdensome and not ideal for many patients

d) PCPs identified potential health care personnel that could assist with SMBP. Although previous studies have indicated that pharmacist can assist with SMBP, PCPs in our study described that additional training may be required to ensure physician confidence. In previous trials where pharmacists have assisted with SMBP, they likely received some additional training. In our interviews, PCPs reiterated the importance of this additional training, which is important for successful implementation. 

Consequently, we consider environmental restructuring and enablement to be the most appropriate strategies to address these barriers. We have included these specific terms in the text to tie back to the conceptual model.

13) Do the authors have a sense for which one of the TDF constructs were the most salient (e.g, are underuse mostly driven by opportunity or motivation and less so capability?).

Response: The ‘environmental context and resources’ seemed the most salient. The respondents spent a lot of time discussing the barriers to SMBP that led to less than desired use. We have added this in our discussion. 

Reviewer#2

1) This manuscript, entitled “Physician perceived barriers and facilitators for self-measured blood pressure monitoring- A qualitative study”, is an important contribution to the literature for self-measured blood pressure monitoring.

Response: Thank you. 

2) Overall, this is a well-written piece, but it could benefit from some additional copy-editing

Response: Thank you for pointing these out. We have corrected these errors as follows: 

a) We have deleted “expected” under “Opportunity” paragraph, third row on page 15 of PDF. 

b) We changed “recoding” to “recording” under “Behavioral Regulation” paragraph, third line on page 13 of PDF. 

c) We changed “somewhat appropriate” to “somewhat inappropriate” in the second to last column of Table 1 on page 22 of PDF. 

d) We changed “HPMB” to “SMPB” in the second column, first entry of Table 2 on page 23 of PDF. 

3) Recommend editing your third paragraph in the background to mention home-blood pressure monitoring. 

Response: We agree. We have clarified that SMBP is also called home blood pressure monitoring in the introduction section. 

4) Could also consider a sentence acknowledging how national initiatives like AMA/AHA’s TargetBP and CDC/CMS’s Million Hearts have been working for years on widespread implementation of SMBP but have not been able to make much headway.

Response: Thank you. We appreciate the suggestion and have added them in the introduction to acknowledge these efforts. 

5) For generalizability purposes, this piece could greatly benefit from some additional contextual information about the respondents and their patient populations. 

a) For example, the authors provide some written demographic data about respondents (e.g. 15/17 women, 3 geriatricians, mix of both academic and community physicians) but it would be nice to have those data and additional data in a table. 

Response: We have added more demographic data of the respondents in the results section. Of the patients with uncontrolled hypertension in the health system, approximately 75% are Caucasian and 18% Black. We expect the respondents to have a similar mix of patients as new patients are randomly assigned physicians in our health system. Obtaining blood pressure control and demographics of specific physicians was outside the scope of our current IRB approval and study design. 

b) Would also be nice to understand if any of these 17 key informants were among the physicians who supported SMBP or were they among those who had reservations about using it (from the survey)? Would also be helpful to have some ballpark estimation of what the respondents actual (and perceived if available) BP control rate is for their patient population, what percent of their population has HTN, etc. 

Response: We agree. We have added Table 1b presenting survey results for the 14 PCPs who completed the interview. While obtaining the actual BP control rates is outside the scope of this study, table 1 a and b indicate similar physician satisfaction and use of SMBP. We have also added overall BP control in the health system. 

c) It would also be helpful to understand a little about the general patient population – e.g. low SES, inner city population; more affluent, mostly-white population. Physician perceptions of barriers and facilitators are skewed by the patients they serve.

Response: We have added details on the general patient population in the subsection “setting” under methods. Evaluating socioeconomic status was beyond the scope of this project as the health system does not enter this information into the EHR. The respondents are expected to have a similar mix of patients as the rest of the health system as in our health system, most new patients are randomly assigned to available physicians. 

6) You mention CPC+. Consider talking more generally about pay-for-quality initiatives because CPC+ is just one of many initiatives in that space. 

Response: We agree. We have changed the sentence to “A few physicians were aware of the health systems participation in pay for quality initiatives such as Comprehensive Primary Care Plus (CPC+), a CMS program, that provides financial incentives for improving primary care delivery, and felt motivated to use SMBP.” to indicate that CPC+ is one of the quality initiatives and our physicians were motivated to use SMBP as our institution participates in CPC+. We specifically mentioned CPC+ as we have it in our health system and the respondents discussed it in their interviews. 

7) You parenthetically mention “A Million Hearts Initiative”. Would recommend changing that to “Million Hearts” and adding more context or a citation there as well. Million Hearts is a major supporter of SMBP but as an initiative, in general, doesn’t pay for monitors. 

Response: Thank you for the comment. We have corrected this error and referenced “Million Hearts” (reference number 23) in the introduction section to acknowledge their efforts in SMBP. 

8) A potential limitation not noted is the fact that almost all respondents were women. While we don’t necessarily have any reason to believe women would have different perceptions about SMBP than men, it is fairly unusual for 88% of respondents in any study to be women and should at least be noted.

Response: We have added this limitation. Interestingly, most of the attendees of the faculty meetings where the initial survey was introduced were women leading to this sex discrepancy. 

9) Another potential limitation is, depending on your response to one of my questions above, the respondents were likely all supporters of SMBP so their barriers are already skewed towards people who trust SMBP. Responses would have differed if respondents included physicians who don’t use SMBP with their patients or who don’t trust the readings, etc.

Response: We have added Table 1b presenting the survey responses from the 14 PCPs who participated in the initial survey and the interviews. Their responses are similar to the responses in Table 1a (from 39 PCPs).

10) I would caution you in the future to separate the use of blood pressure kiosks (i.e. found in pharmacies/grocery stores) from SMBP performed on a patient’s own blood pressure monitor. Not a lot of evidence behind the former, lots behind the latter. If you discussed both of them equivalently with physicians in your interviews (the way it was denoted in the survey), that may have skewed their perceptions. May want to list this as a potential limitation.

Response: We agree. We were trying to limit the number of questions in the initial survey and thus ended up combining these as out of office BP in one of the questions. For the interviews we used the term ‘home BP monitoring’ (that did not include use of blood pressure kiosks) (supplementary data 1). 

11) Future research is not noted but there seemed to be some gaps in in what was asked of these physicians. For example, what were their feelings about blood pressure device loaner programs (where health care settings bulk purchase devices and loan them out to patients on a temporary basis)? Were they aware of the new CPT codes that were introduced in 2020 that would reimburse clinicians one time to train patients on SMBP technique/perform device calibration (99473) and reimburse physicians up to monthly to receive SMBP readings, interpret, and combine into a patient’s care plan (99474)?

Response: We agree that these would add to the information collected. Since we started the interviews in late 2019 - early 2020, we did not ask about the new CPT codes. None of the respondents mentioned them either. Our institution does not have a BP loaner program and thus we did not ask about that either. We have added these to our limitations. 

12) References 30 and 46 appear to be the same reference.

Response: Thank you for pointing this out. We have corrected the references. 

Other changes: 

Minor changes in the abstract 

We found some errors in Table and have corrected them.

Additional changes in discussion to decrease the overall word count 

We replaced Figure 1 due to a typo in the original figure

---

## [Decision Letter · Decision Letter 1]

21 Jun 2021

PONE-D-20-38491R1

Physician perceived barriers and facilitators for self-measured blood pressure monitoring- A qualitative study

PLOS ONE

Dear Dr. Gupta,

Thank you for submitting your manuscript to PLOS ONE. After careful consideration, we feel that it has merit but does not fully meet PLOS ONE’s publication criteria as it currently stands. Therefore, we invite you to submit a revised version of the manuscript that addresses the points raised during the review process.

ACADEMIC EDITOR:

Please incorporate the second reviewer's minor comments. 

Thank you.

regards,

Vijay. 

We look forward to receiving your revised manuscript.

Kind regards,

Vijayaprakash Suppiah, PhD

Academic Editor

PLOS ONE

Journal Requirements:

Reviewers' comments:

Reviewer's Responses to Questions

**Comments to the Author**

1. If the authors have adequately addressed your comments raised in a previous round of review and you feel that this manuscript is now acceptable for publication, you may indicate that here to bypass the “Comments to the Author” section, enter your conflict of interest statement in the “Confidential to Editor” section, and submit your "Accept" recommendation.

Reviewer #1: All comments have been addressed

Reviewer #2: All comments have been addressed

2. Is the manuscript technically sound, and do the data support the conclusions?

Reviewer #1: Yes

Reviewer #2: (No Response)

3. Has the statistical analysis been performed appropriately and rigorously? 

Reviewer #1: Yes

Reviewer #2: (No Response)

4. Have the authors made all data underlying the findings in their manuscript fully available?

Reviewer #1: Yes

Reviewer #2: (No Response)

5. Is the manuscript presented in an intelligible fashion and written in standard English?

Reviewer #1: Yes

Reviewer #2: (No Response)

6. Review Comments to the Author

Reviewer #1: (No Response)

Reviewer #2: This manuscript, entitled “Physician perceived barriers and facilitators for self-measured blood pressure monitoring- A qualitative study”, is an important contribution to the literature for self-measured blood pressure monitoring. The authors have done a nice job addressing previous reviewer comments.

A few minor comments:

1. Suggest reviewing it again for grammar editing – found multiple punctuation (e.g. inconsistent use of Oxford comma), spelling (e.g. “specifc”, “fulfil”), and grammar mistakes (e.g. missing hyphens in compound adjectives like “patient care related”, “evidence based”). Proper grammar and editing will add needed clarity throughout.

2. Suggest editing the sentence “A majority of patients are Caucasian, 6% Hispanic, 2.5% Asian, 1.7% Black, and 18% unknown or other.” to “The racial/ethnic breakdown of patients was XX% Caucasian, 6% Hispanic, 2.5% Asian, 1.7% Black, and 18% unknown or other.

3. “Approximately 49% of all patients with a BP reading had atleast…” – need a space in ‘at least’.

4. You use both “White” and “Caucasian” in describing race in your study. Suggest picking one (White).

5. Under the Opportunity section, the authors state “Physicians saw an opportunity for SMBP. Under Social Influences, most physicians identified published evidence as the major reason for using SMBP, and not peer influence. Since SMBP is backed by evidence, some PCPs found it concerning if others did not use SMBP. Only a change in evidence, and not the behavior of their colleagues, would make them not use SMBP (Table 3, social influences).” The finding that is more interesting than what would make SMBP users change their practice is what would make SMBP non-users change their practice. The way this paragraph is written, there is no opportunity on which we can act.

6. “Under Behavioral Regulation respondents elaborated differences in using SMBP for managing HTN.” Need the word “on” after “elaborated”.

7. “Despite their beliefs in the strong evidence behind SMBP, there was underutilization of SMBP due to logistical barriers, lack of resources and trained personnel to help with SMBP.” – need an “and” before “lack”

8. “…suggesting need for better strategies to monitor SMBP” – need the word “the” before “need”

9. “(such as use of patient portal for entering SMBP)” – need “the” or “a” before “patient portal”

10. “performance assessments, standardized protocols” – need an “and” after the comma

11. “With stakeholders; patients, physicians, health systems and insurers interested in adopting SMBP, practical strategies that are scalable are needed.” – this is an awkward sentence. Suggest rewrite of “With stakeholders such as patients, physicians, health systems, and insurers interested in adopting SMBP, practical, scalable strategies are needed.”

12. Table 2 says SMPB instead of SMBP

7. PLOS authors have the option to publish the peer review history of their article (what does this mean?). If published, this will include your full peer review and any attached files.

Reviewer #1: **Yes: **Nathalie Moise

Reviewer #2: No

---

## [Author Response · Author response to Decision Letter 1]

15 Jul 2021

We thank the editors and the reviewers for their thoughtful feedback. Our responses to the concerns are outlined below. 

Reviewer #2

1) This manuscript, entitled “Physician perceived barriers and facilitators for self-measured blood pressure monitoring- A qualitative study”, is an important contribution to the literature for self-measured blood pressure monitoring. The authors have done a nice job addressing previous reviewer comments.

Response: Thank you. 

2) Suggest reviewing it again for grammar editing. 

Response: Thank you for pointing out errors, typos and unclear sentences. We have reviewed the entire manuscript again and corrected these errors according to your suggestions. 

3) Under the Opportunity section, the authors state “Physicians saw an opportunity for SMBP. Under Social Influences, most physicians identified published evidence as the major reason for using SMBP, and not peer influence. Since SMBP is backed by evidence, some PCPs found it concerning if others did not use SMBP. Only a change in evidence, and not the behavior of their colleagues, would make them not use SMBP (Table 3, social influences).” The finding that is more interesting than what would make SMBP users change their practice is what would make SMBP non-users change their practice. The way this paragraph is written, there is no opportunity on which we can act.

Response: We agree. We have rephrased this section. 

Other changes: 

We added some more demographic details on primary care providers (requested in the previous review) which we were now able to obtain from our health system. 

We removed the abbreviation ‘HTN’ for hypertension.

---

## [Editor Report · Decision Letter 2]

21 Jul 2021

Physician perceived barriers and facilitators for self-measured blood pressure monitoring- A qualitative study

PONE-D-20-38491R2

Dear Dr. Gupta,

We’re pleased to inform you that your manuscript has been judged scientifically suitable for publication and will be formally accepted for publication once it meets all outstanding technical requirements.

Kind regards,

Vijayaprakash Suppiah, PhD

Academic Editor

PLOS ONE

---

## [Editor Report · Acceptance letter]

12 Aug 2021

PONE-D-20-38491R2 

Physician perceived barriers and facilitators for self-measured blood pressure monitoring- A qualitative study 

Dear Dr. Gupta:

I'm pleased to inform you that your manuscript has been deemed suitable for publication in PLOS ONE. Congratulations! Your manuscript is now with our production department. 

Kind regards, 

on behalf of

Dr. Vijayaprakash Suppiah 

Academic Editor

PLOS ONE